# Combustion Kinetics Characteristics of Solid Fuel in the Sintering Process

**Jihui Liu [1], Yaqiang Yuan [1], Junhong Zhang [1], Zhijun He [1],\* and Yaowei Yu [2],\***

[1]   School of Materials and Metallurgy, University of Science and Technology, Liaoning 114051, China; gtyj66@126.com (J.L.); as2013h@163.com (Y.Y.); gtyj0412@163.com (J.Z.)
[2]   State Key Laboratory of Advanced Special Steel, School of Materials Science and Engineering, Shanghai University, Shanghai 200444, China
\*   Correspondence: hezhijun@ustl.edu.cn (Z.H.); yaoweiyu@shu.edu.cn (Y.Y.)

**Abstract:** In order to systematically elucidate the combustion performance of fuel during sintering, this paper explores the influence of three factors, namely coal substitution for coke, quasi-particle structure and the coupling effect with reduction and oxidation of iron oxide, on fuel combustion characteristics, and carries out the kinetic calculation of monomer blended fuel (MBF) and quasi-granular fuel (QPF). The results show that replacing coke powder with anthracite can accelerate the whole combustion process. MBF and QPF are more consistent with the combustion law of the double-parallel random pore model. Although the quasi-particle structure increases the apparent activation energy of fuel combustion, it can also produce a heat storage effect on fuel particles, improve their combustion performance, and reduce the adverse effect of diffusion on the reaction process. In the early stage of reaction, the coupling between combustion of volatiles and reduction of iron oxide is obvious. The oxidation of iron oxide will occur again when the combustion reaction of fuel is weakened.

**Keywords:** quasi-particle structure; monomer blended fuel; quasi-particle fuel; apparent activation energy; coupling effect

---

## 1. Introduction

At present, the supply of energy is highly reliant on fossil fuel [1]. The rising demand for energy around the globe is leading to many economic and environmental problems [2]. Ironmaking in China is still dependent on the blast furnace [3]. The direct reducibility and intensity of high-basicity sinter, which is the main raw material for blast furnace ironmaking, markedly influence the production efficiency of ironmaking [4,5]. The combustion characteristic of fuel in sintering mixture materials plays a decisive role in sinter quality [6,7]. Therefore, it is urgent to explore the combustion characteristics of solid fuel in the sintering process in order to provide a theoretical basis for improving the combustion efficiency, improving the quality of sinter and reducing the consumption of solid fuel [8]. Coke is made from natural bituminous coal heated between 950 °C and 1050 °C in an airless environment. It is the main fuel in the sintering site [9]. Previously, the research on sintering fuel combustion was mostly based on coke breeze [10,11]. However, with the development of the steel industry and the continuous improvement of the grog ratio in the blast furnace, the supply of coke powder for sintering has been short. Therefore, many sintering sites use anthracite wholly or partially to replace coke powder as fuel for sintering [12,13].

In the traditional sense, the experimental samples for studying the combustion characteristics of fuels consist mainly of monomer fuel. However, during the sintering process, the solid fuel in the material layer is usually distributed in a dispersed manner. Hence, the combustion law of sintering fuel should generally be different from the monomer fuel and the fuel layer [14,15]. In recent years,

researchers have begun to use a quasi-particle structure to describe the existence of fuel inside the sintering mixture [16,17]. In order to exclude the interference brought by other reactions, $Al_2O_3$ pure powder reagent is often used in many studies on quasi-particles sinter to replace other materials used in production for experimental exploration [18].

Combustion of sintering fuel at low temperature is generally considered as a chemical reaction control process. However, the introduction of a quasi-particle structure will greatly improve the diffusion resistance of internal fuel combustion. Therefore, the extent to which diffusion controls combustion is greatly increased [19,20]. This paper innovatively introduced the double parallel reaction volume model (DVM) and double parallel random pore model (DRPM) to conduct a comparative analysis of the combustion characteristics of monomer blended fuel and quasi-particle fuel and calculated the related kinetic parameters, which are widely used in the kinetic calculation of co-combustion of multiple fuels [21], single-fuel gasification [22] and co-gasification of multiple fuels [23,24]. They can not only obtain the suitable dynamic models for describing the combustion process of two kinds of fuel, but also characterize the effect of quasi-particle structure on the combustion characteristics of sintering fuel. There are many physical and chemical reactions in the sintering process. Due to the influence of sintering temperature and atmosphere, iron oxides undergo different degrees of reduction and oxidation reactions. The occurrence of these reactions is coupled with fuel combustion, which greatly affects the quality of sinter [25]. Therefore, this study systematically explores the influence of factors such as the substitution of anthracite for coke powder, quasi-particle structure and the coupling effect of reduction and oxidation of iron oxide on the combustion characteristics of sintering fuel in order to provide a certain guiding significance for improving fuel efficiency and reducing sintering production cost.

## 2. Materials and Methods

### 2.1. Materials

The fuel used in the experiment was 25% anthracite blended with 75% coke powder by weight and the particle size was less than 0.105 mm. The specific industrial analysis, elemental analysis and calorific value of the raw materials are shown in Table 1. Because there are many types of reactions of the sintering process of iron ore and the coupling between the reactions is strong, in order to investigate the influence of the quasi-particle structure on the combustion kinetic parameters of the sintering fuel, $Fe_2O_3$ and $Al_2O_3$ pure powder reagents with a particle size of less than 0.147 mm were used instead of the iron ore and flux applied in an industrial setting. In addition to single anthracite and coke powder, there are also three-blended fuels, namely monomer blended fuel (MBF), quasi-particle fuel (QPF) and sintered mixture (SDM). The schematic diagrams of them are shown in Figure 1, and the ratios of raw materials are shown in Table 2.

**Table 1.** Proximate and ultimate analysis of the fuel (in dry basis).

| Samples | Proximate Analysis/% | | | Ultimate Analysis/% | | | | | $Q_{gr}/(MJ\ g^{-1})$ |
|---|---|---|---|---|---|---|---|---|---|
| | **FC** | **V** | **A** | **C** | **H** | **O** | **N** | **S** | |
| **Anthracite** | 76.51 | 6.94 | 16.58 | 75.86 | 1.69 | 2.25 | 0.81 | 0.36 | 29.31 |
| **Coke** | 86.68 | 0.23 | 11.80 | 87.17 | 0.77 | 0.64 | 0.91 | 0.51 | 35.23 |

FC, V and A represent fixed carbon, volatile and ash, respectively; subscript gr means gross calorific value.

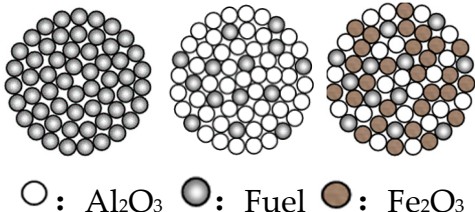

○ ： Al₂O₃　○ ： Fuel　● ： Fe₂O₃

**Figure 1.** Schematic diagram of monomer blended fuel (MBF), quasi-granular fuel (QPF) and sintered mixture (SDM).

**Table 2.** Raw materials and proportions used in the experiment, wt%.

| Samples | $Fe_2O_3$ | $Al_2O_3$ | Anthracite | Coke |
|---------|-----------|-----------|------------|------|
| **MBF** | - | - | 25 | 75 |
| **QPF** | - | 60 | 10 | 30 |
| **SDM** | 76 | 20 | 1 | 3 |

*2.2. Thermogravimetric Experiment*

Combustion experiments were conducted using the HCT-4 thermal analyzer. The usage of anthracite, coke powder and MBF was $1.2 \pm 0.1$ mg for each group, and $30 \pm 0.1$ mg for QPF and SDM respectively. Each sample was loaded into a crucible for thermal analysis with a height of 4 mm and diameter of 5 mm. The samples were heated from room temperature (25 °C) to 1000 °C at heating rates of 5.0, 10.0, 15.0, and 20.0 °C/min, respectively. The rate of heating is expressed as $\beta$. At the same time, the air was injected at a flow rate of 100 mL/min during the heating to provide an oxidizing atmosphere for the heating. To ensure the accuracy of all experimental results, each experiment was repeated at least three times under the same conditions.

The conversion of the sample ($\alpha$) was calculated with the mass loss data collected during the combustion

$$\alpha = \frac{m_0 - m_t}{m_0 - m_\infty} \tag{1}$$

where $m_0$ is the original mass of the sample; $m_t$ is the mass at time t; $m_\infty$ is the final mass of the sample after the reaction.

In the thermogravimetric combustion experiment, the parameters of the sample can be determined by using the thermal analysis curve (TG-DTG), including ignition temperature ($T_i$), combustion temperature ($T_j$), peak temperature ($T_p$), combustion reaction time ($t$), flammability index ($C$) and combustion characteristic index ($S$). The determination method of characteristic parameters is shown in Figure 2.

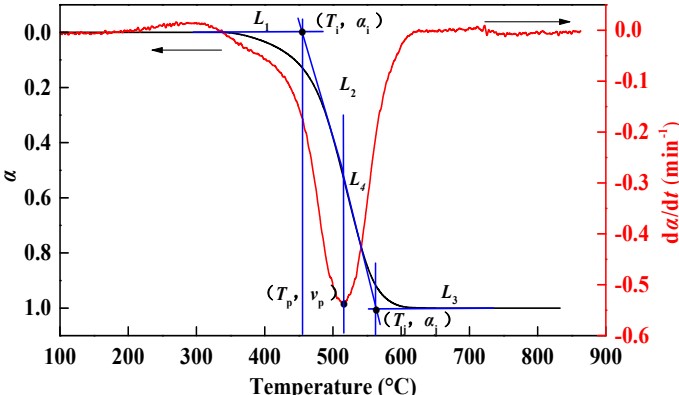

**Figure 2.** Schematic diagram of characteristic parameter determination method of the thermogravimetric curve.

The flammability index reflects the ability of the sample to react at the beginning of combustion. This index can measure the ignition stability of the sample during combustion,

$$C = v_p / T_i^2 \tag{2}$$

where $v_p$ is the maximum combustion reaction rate in $\text{min}^{-1}$.

The combustion characteristic index reflects a combined characteristic of the ignition and combustion of the sample. If the value of $S$ is larger, the combustion performance of the sample is better,

$$S = v_p \times v_m / (T_i^2 \times T_j) \tag{3}$$

where $v_m$ is the average burning rate of the sample from $T_i$ to $T_j$ in $\text{min}^{-1}$.

### 2.3. Thermal Analysis Kinetic

The combustion process of the sintering fuel can be regarded as a gas–solid heterogeneous reaction. The total combustion reaction consists of two independent chemical reactions:

$$\text{Anthracite} + (x/2 + y + z/2)\text{O}_2 \rightarrow x\text{CO} + y\text{CO}_2 + z\text{H}_2\text{O} \tag{4}$$

$$\text{Coke} + (x/2 + y + z/2)\text{O}_2 \rightarrow x\text{CO} + y\text{CO}_2 + z\text{H}_2\text{O} \tag{5}$$

In order to further clarify the combustion reaction mechanism of the pure mixed fuel and the quasi-particle fuel, we introduced two kinetic models to study the combustion behavior of the sample:

$$\frac{\text{d}\alpha}{\text{d}t} = \sum_{i=1}^{2} c_i k_i f(\alpha_i) \tag{6}$$

where $t$ is the reaction time, $c_i$ is the proportion of a reaction to the total response, $k_i$ is the combustion reaction rate constant, and $f(\alpha_i)$ is a function of the differential reaction mechanism.

The relationship between apparent reaction rate and temperature can be derived from the Arrhenius equation,

$$k = Ae^{-E/RT} \tag{7}$$

where $E$ is reaction activation energy, $A$ is the pre-exponential factor, $R$ is the universal gas constant, and $T$ is the temperature.

Currently, volumetric models (VM) and random pore models (RPM) are widely used to describe various coal char combustion reactions and are used to calculate kinetic parameters,

$$\frac{d\alpha_{\text{VM}}}{dt} = A_{\text{VM}} e^{-E_{\text{VM}}/RT} (1 - \alpha_{\text{VM}}) \tag{8}$$

$$\frac{d\alpha_{\text{RPM}}}{dt} = A_{\text{RPM}} e^{-E_{\text{RPM}}/RT} (1 - \alpha_{\text{RPM}}) \sqrt{1 - \psi \ln(1 - \alpha_{\text{RPM}})} \tag{9}$$

where $\psi$ is the parameter of particle structure,

$$\psi = \frac{4\pi L_0 (1 - \varepsilon_0)}{S_0^2} \tag{10}$$

where $S_0$ is the pore surface area, $L_0$ is the pore length, and $\varepsilon_0$ is the porosity of particles.

Since the experimental materials use two types of fuels with different combustion performances, it is necessary to optimize the VM and RPM. The expressions of DVM and DRPM are obtained by combining Equations (6)–(9),

$$\frac{d\alpha_{DVM}}{dt} = \sum_{i=1}^{2} c_i A_i e^{-Ei/RT} (1 - \alpha_i) \tag{11}$$

$$\frac{d\alpha_{DRPM}}{dt} = \sum_{i=1}^{2} c_i A_i e^{-E_i/RT} (1 - \alpha_i) \sqrt{1 - \psi_i \ln(1 - \alpha_i)} \tag{12}$$

In the non-isothermal analysis experiment, in order to determine the kinetic parameters and improve the calculation accuracy, three or more types of heating rates are usually selected. Thus, this experiment adopts four different heating rates to calculate kinetic parameters. Under the constant heating rate of the experiment, the reaction temperature can be obtained from the initial temperature and reaction time,

$$T = T_0 + \beta t \tag{13}$$

where $\beta$ is the heating rate, and $T_0$ is the starting temperature of 25 °C. After $t = (T - T_0)/\beta$ is substituted into Equations (11) and (12), the formulas can be integrated to give

$$\alpha_{DVM} = \sum_{i=1}^{2} c_i \left( 1 - \exp\left( -\frac{A_i R T^2}{\beta E_i} \cdot \exp\left( \frac{-E_i}{RT} \right) \right) \right) \tag{14}$$

$$\alpha_{DRPM} = \sum_{i=1}^{2} c_i \left( 1 - \exp\left( -\exp\left( \frac{-E_i}{RT} \right) \cdot \frac{A_i R T^2}{\beta E_i} \cdot \left( 1 + \exp\left( \frac{-E_i}{RT} \right) \cdot \frac{\psi_i A_i R T^2}{4\beta E_i} \right) \right) \right) \tag{15}$$

The combustion kinetic parameters were calculated by the above two kinetic models at different heating rates. The experimental data of the reaction rate ($d\alpha/dt$) and conversion rate ($\alpha$) were fitted in 1stop software using a nonlinear least-squares method. Then, the parameters $A$, $E$ and $\psi$ that are obtained are substituted into Equations (14) and (15) to obtain the relationship between the sample conversion rate ($\alpha$) and the temperature ($T$) during combustion. At the same time, due to the possible deviation between the actual value and the calculated value of the model, the root mean square error (*RMSE*) is introduced to evaluate the error between the fitted data and the actual value of the DVM and DRPM models,

$$RMSE(\alpha) = \frac{\sqrt{\sum_{i=1}^{N} \left( \alpha_{exp}^i - \alpha_{cal}^i \right)^2}}{N} \times 100\% \tag{16}$$

$$RMSE\left( \frac{d\alpha}{dt} \right) = \frac{\sqrt{\sum_{i=1}^{N} \left( \frac{d\alpha}{dt}_{exp}^i - \frac{d\alpha}{dt}_{cal}^i \right)^2}}{N} \times 100\% \tag{17}$$

where $\alpha_{exp}^i$ and $\alpha_{cal}^i$ are the experimental and calculated values of the conversion rate at points $i$ = 1, 2, 3, ... ; $\frac{d\alpha}{dt}_{exp}^i$ and $\frac{d\alpha}{dt}_{cal}^i$ are the experimental and calculated values of reaction rate at some points, and $N$ is the number of data points.

## 3. Results and Discussion

### 3.1. FTIR Analysis

In order to compare and analyze the combustion characteristics of anthracite and coke powder used in sintering site, this experiment adopts Fourier transform infrared spectroscopy to detect the functional group structure of the two fuels and compare the differences between the structures, thus providing a theoretical basis for studying their combustion characteristics and laws. In infrared spectrum detection, the absorption peak of each functional group has a specific spectral position.

The corresponding functional group can be found according to the peak of the characteristic peak on the spectrum. According to relevant studies [26,27], the infrared spectral curves of coal samples can be divided into hydroxyl (3700~3000 cm$^{-1}$), aliphatic hydrocarbon (3000~2800 cm$^{-1}$), oxygen-containing functional group (1800~1000 cm$^{-1}$) and aromatic hydrocarbon (900~700 cm$^{-1}$) according to the spectral wave number, the structure and properties of functional groups. Therefore, in this paper, the infrared spectral curves of anthracite and coke powder were divided into nine points A~I in order to better distinguish the differences in the structure of their functional groups. It can be seen from the Figure 3 that there is a big gap between the two spectral curves.

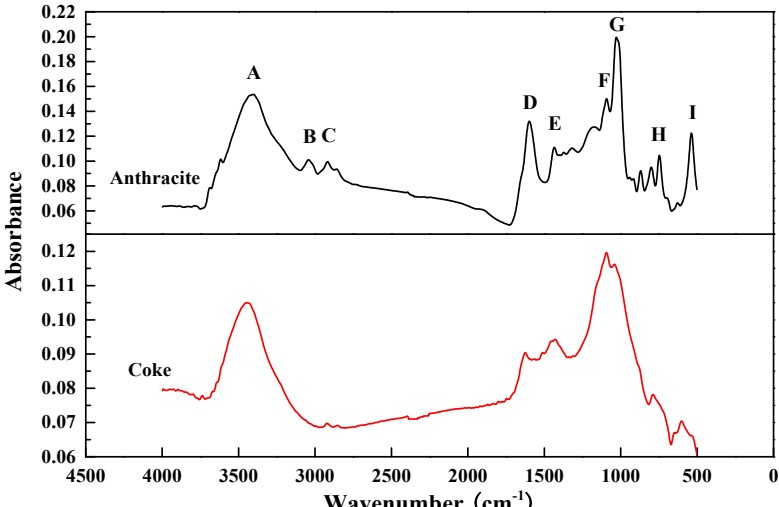

**Figure 3.** Fourier infrared spectra of anthracite and coke.

It can be seen from Table 3 that there are significant differences between coke powder and anthracite at B and H, which represent the –CH group on aromatic ring and aliphatic hydrocarbon in coal respectively. As the content of –CH will increase with the increase of volatile substances, compared with coke powder, anthracite still contains a small amount of volatile substances, so the content of –CH group in anthracite is significantly higher than that of coke powder. At the same time, the comparison of the absorbance of the two at point A in the infrared spectrum shows that the content of hydroxyl in the pyrolysis process of coking coal is greatly reduced, that is, the number of active groups and the activity of coal are reduced, so the reactivity of coke powder is significantly lower than that of anthracite. This is also the reason why the mixed combustion process of anthracite and coke powder for sintering will present a multi-stage weightless reaction.

**Table 3.** Infrared spectrum absorption peak classification of anthracite and coke powder.

| Peak Position | Wavenumber/cm$^{-1}$ | | Functional Group |
|:---:|:---:|:---:|:---:|
| | Anthracite | Coke | |
| A | 3414.4 | 3439.1 | –OH |
| B | 3042.2 | - | –CH (Aromatic hydrocarbon) |
| C | 2923.3 | 2914.7 | –CH$_3$, –CH$_2$– |
| D | 1604.1 | 1626.3 | –C=C– |
| E | 1439.1 | 1444.1 | –CH$_2$– |
| F | 1089.9 | 1094.2 | C–O (Phenol, alcohol, ether, ester) |
| G | 1023.2 | 1037.7 | –Si–O– |
| | 870.1 | - | Carbonate minerals |
| H | 793.1 | 786.2 | Substituted benzene class |
| | 746.9 | - | –CH$_2$– |
| I | 542.3 | 540.6 | –S–S– |

### 3.2. Thermogravimetric Characteristics

#### 3.2.1. Combustion Characteristics of Anthracite and Coke

In order to better characterize the combustion process of anthracite and coke powder for sintering, this study conducted a TG-DTG analysis on the two fuels by means of thermogravimetric experiments at the same heating rate. It can be seen from Figure 4 that the combustion interval of anthracite and coke powder is concentrated in a certain region, and there is only one section of weightlessness. Because the volatile content of anthracite and coke powder is low, most of the reaction weight loss can be considered as the result of fixed carbon combustion. However, the volatile content of anthracite is 6.94%, which is significantly higher than that of coke powder. Therefore, the combustion interval of anthracite is obviously smaller than that of coke powder.

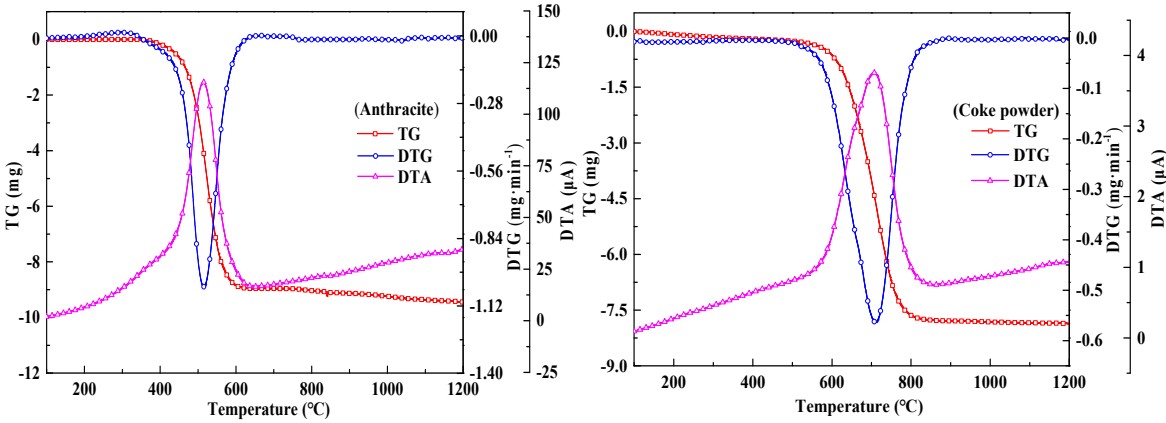

**Figure 4.** Thermogravimetric curves of anthracite and coke.

Combined with Equations (2) and (3), the combustion characteristic parameters of anthracite and coke powder for sintering can be obtained. As shown in Table 4, the average combustion rate, ignition stability $C$ value and combustion characteristic index $S$ value of anthracite were higher than that of coke powder, while the combustion time was lower than that of coke powder. Therefore, anthracite has a better combustion performance than coke powder. This is because the volatile content of coke powder and the strength of hydroxyl absorption peak are low, which directly leads to the ignition temperature of anthracite being lower than that of coke powder, resulting in significant differences in the combustion performance of the two. Therefore, the coal coke mixed fuel used in sintering raw materials will have multi-stage weightless reactions during the combustion process.

**Table 4.** Combustion characteristics of anthracite and coke powder.

| Samples | $T_i/(°C)$ | $T_p/(°C)$ | $v_p/(min^{-1})$ | $T_j/(°C)$ | $v_m/(min^{-1})$ | $C \times$ $10^{-6}/(min^{-1}·°C^{-2})$ | $S \times$ $10^{-9}/(min^{-2}·°C^{-3})$ | $t/(min)$ |
|---|---|---|---|---|---|---|---|---|
| **Anthracite** | 353.7 | 516.6 | 1.0529 | 619.3 | 0.0974 | 8.4562 | 1.3237 | 10.27 |
| **Coke** | 443.5 | 708.7 | 0.5672 | 846.4 | 0.0726 | 2.8837 | 0.2473 | 13.77 |

#### 3.2.2. Combustion Characteristics of MBF and QPF

Figure 5 shows the conversion ($\alpha$) and the reaction rate ($d\alpha/dt$) of MBF and QPF at different heating rates. The improving trend of the reaction rates gradually slows down as the heating rate increases and all the reaction curves have a common feature that the weight loss reaction of the two samples is divided into two parts, which is because that the different combustion characteristics of anthracite and coke divide the whole combustion process into two reaction zones. The first reaction zone is mainly the combustion reaction of anthracite, which occurs in the temperature range of about 400–600 °C. The second reaction zone is the combustion of coke breeze, and the temperature range

of this reaction is about 550–950 °C. Because of the difference in the amount of anthracite and coke powder added, the rate of change peak of the latter is significantly higher than that of the former.

As shown in Table 5, when the heating rate ($\beta$) is increased from 5 °C/min to 20 °C/min, $T_i$, $T_j$, $T_{p-1}$ and $T_{p-2}$ all increase, and the conversion rate and reaction rate curves are shifted to the high-temperature region, which indicates that the amount of heat transferred from the surrounding environment to the inside of the sample per unit time increases. The increase of the heat will greatly improve the combustion rate of the fuel but also shortens the reaction time ($t$) of samples at the same temperature, so the phenomenon of the curve's movement to the high-temperature area will occur. By comparing the $T_{p-2}$ and $T_j$ values of MBF in Table 5 at 5 °C/min with the $T_p$ and $T_j$ values in Table 4, it can be seen that the maximum reaction rate and the corresponding temperature at the end reaction of the coke powder in the mixed fuel are significantly lower than the corresponding temperature when the coke powder is burned alone. This indicates that the combustion of anthracite before coke powder provides heat for the latter's oxidation, thus speeding up the reaction process of coke powder. Therefore, when the sintering site adopts anthracite to partially replace coke powder, the migration speed of combustion zone will be accelerated and the sintering efficiency will be improved.

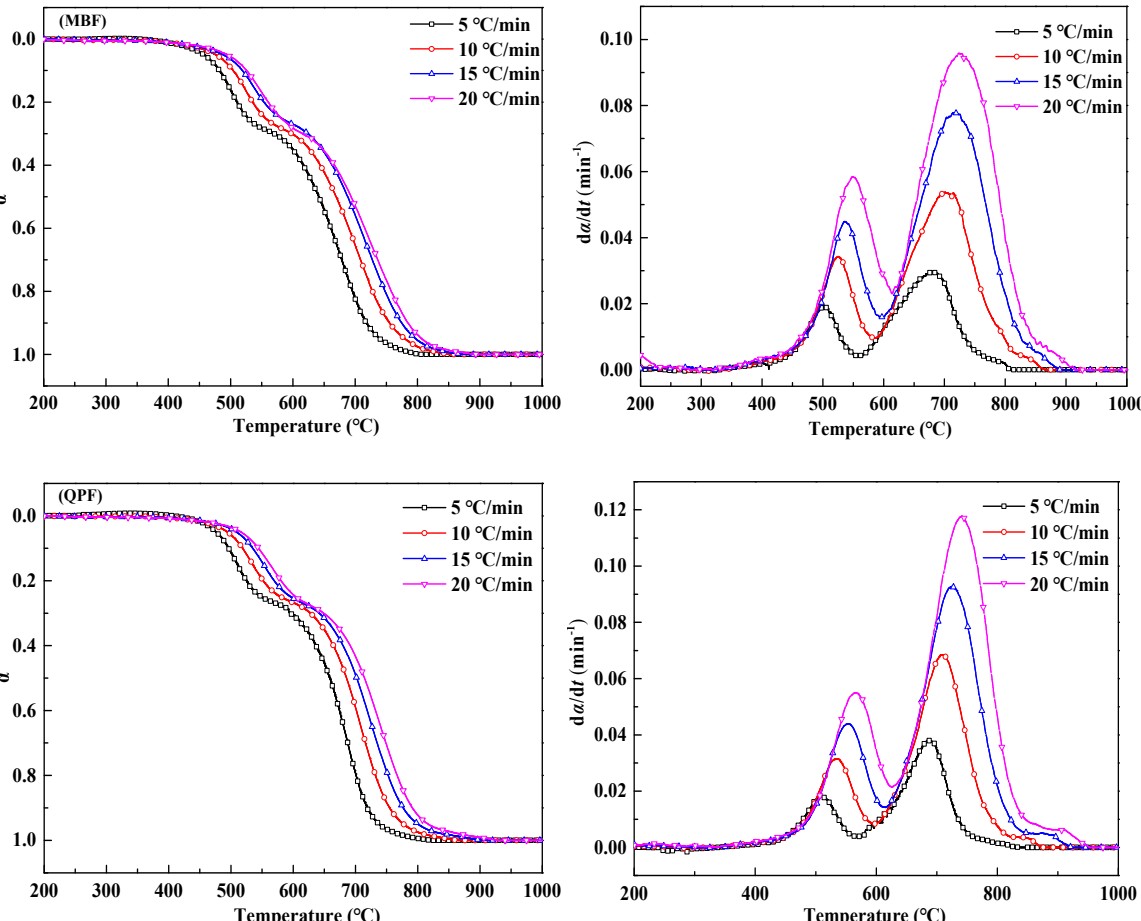

**Figure 5.** Fractional conversion and reaction rate-conversion curves of MBF and QPF at different heating rates.

**Table 5.** Combustion characteristic parameters of MBF and QPF at different heating rates.

| Sample | $\beta/(°C\ min^{-1})$ | $T_i$ (°C) | $T_{p-1}$ (°C) | $v_{p-1}$ (min$^{-1}$) | $T_{p-2}$ (°C) | $v_{p-2}$ (min$^{-1}$) | $T_j$ (°C) | $v_m$ (min$^{-1}$) | $C \times 10^{-7}$ (min$^{-1}$·°C$^{-2}$) | $S \times 10^{-12}$ (min$^{-2}$·°C$^{-3}$) | $t$ (min) |
|---|---|---|---|---|---|---|---|---|---|---|---|
| MBF | 5 | 399.2 | 501.2 | 0.020 | 676.1 | 0.029 | 802.8 | 0.012 | 1.63 | 2.52 | 60.32 |
| | 10 | 423.1 | 523.2 | 0.034 | 698.6 | 0.054 | 856.1 | 0.023 | 2.62 | 7.07 | 33.29 |
| | 15 | 419.4 | 536.3 | 0.045 | 717.3 | 0.078 | 876.9 | 0.033 | 3.79 | 14.18 | 22.71 |
| | 20 | 436.0 | 543.3 | 0.057 | 717.3 | 0.094 | 895.1 | 0.044 | 4.32 | 20.05 | 17.59 |
| QPF | 5 | 414..3 | 509.0 | 0.018 | 681.9 | 0.038 | 826.1 | 0.012 | 1.86 | 2.73 | 63.42 |
| | 10 | 432.6 | 531.5 | 0.031 | 707.4 | 0.068 | 838.6 | 0.025 | 3.06 | 8.99 | 30.71 |
| | 15 | 447.8 | 549.9 | 0.044 | 719.9 | 0.092 | 901.1 | 0.033 | 3.89 | 14.29 | 23.41 |
| | 20 | 456.8 | 561.4 | 0.055 | 732.2 | 0.116 | 917.5 | 0.043 | 5.29 | 24.81 | 17.81 |

It can be seen that the QPF flammability index, combustion characteristic index and combustion reaction time are significantly higher than MBF, while the average burning rate of both is about the same. This indicates that the quasi-particle structure extends the reaction time of fuel combustion, but it does not inhibit the burning rate and in fact, improves the combustion performance. It is probable that the inert particles around the fuel reduce the loss of heat generated by the combustion reaction. At the same time, as the combustion reaction progresses, the particle size of the fuel gradually shrinks, so that a "regenerator" is formed around the wrapped fuel. The specific schematic diagram is shown in Figure 6. This is also the reason for the heat storage in the combustion zone during sintering.

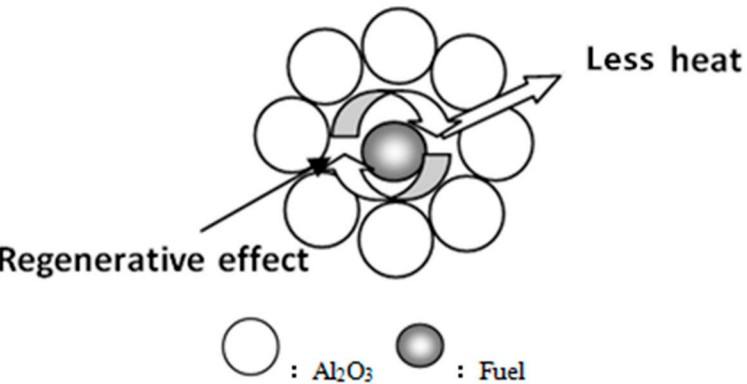

**Figure 6.** Thermal storage of quasi-particle sintering.

### 3.3. Combustion Kinetic Parameters

The relationship between the conversion rates and reaction rates of the two kinds of fuel at different heating rates is shown in Figure 7. The whole combustion reaction process is divided into two stages. The earlier stage is mainly anthracite combustion, while the latter stage is mainly composed of coke-powder combustion. The trend of the curves describing the two combustion stages is roughly the same. The rate in the initial stage of the reaction increases rapidly with an increase in the conversion rate and then decreases rapidly after reaching a certain peak value. The combustion reaction of anthracite and coke breeze follow the law of non-uniform reaction. Therefore, the combustion reaction of the fuel in the low-temperature condition is in the "chemical-controlled zone", where the combustion rate is greatly affected by the temperature, and the reaction rate increases with an increase of temperature; As the reaction proceeds, the ash content on the fuel surface gradually increases and adheres to the particle surface, which limits the diffusion of gas to the solid boundary layer and the desorption of gaseous reaction products from the solid surface to a certain extent, then greatly slows down the rate of late combustion reactions. After the peak, the reaction enters the "diffusion-controlled zone". Because of the dependence of reaction rate on the diffusion rate of the gas, the burning rate of the fuel decreases as the conversion rate increases.

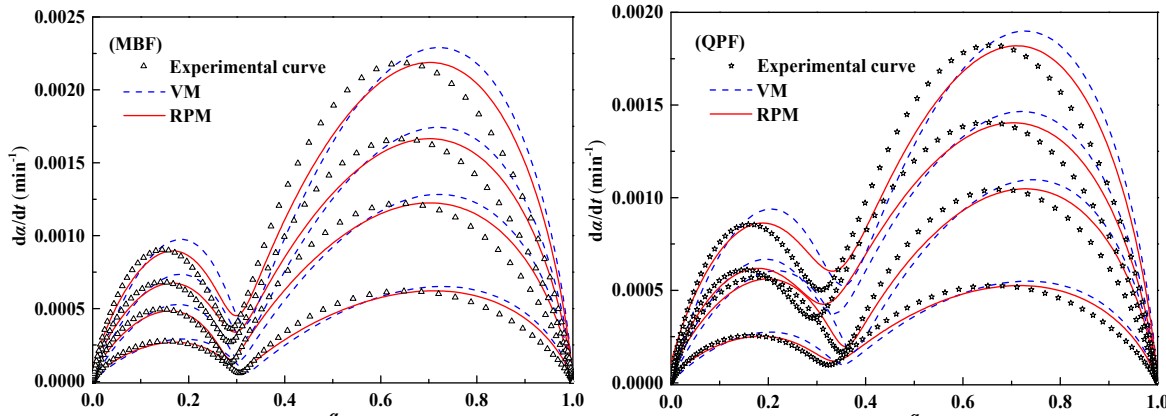

**Figure 7.** Combustion rates of MBF, QPF and fitting curves of double parallel reaction volume model (DVM) and double parallel random pore model (DRPM).

Table 6 lists the kinetic parameters and correlation coefficient $R^2$ of MBF and QPF calculated according to Equations (14) and (15). It can be seen from the data that the correlation coefficients of the two samples calculated by the DVM model are all ≤ 0.9993, and the correlation coefficients calculated by DRPM are all ≥ 0.9994. Whether it is MBF or QPF, the correlation coefficient of the data calculated by the latter is slightly higher than by the former. Table 7 lists the *RMSEs* of conversion and combustion rates calculated according to Equations (16) and (17). The results show that the *RMSEs* of conversion and combustion rate calculated by DRPM model at different heating rates are less than 0.8 and 0.01, respectively, which are far less than the *RMSEs* of the conversion rate and combustion rate calculated by the DVM model. Therefore, the combustion behavior of the MBF and QPF follows the combustion law of the double parallel reaction random pore model.

To further verify the reliability of the data calculated by the DRPM model, the corresponding kinetic parameters in Table 6 were brought into Equation (15), and the resulting conversion curves were compared with the experimental curves. As shown in Figure 8, $\alpha_1$ and $\alpha_2$ increase with temperature at different heating rates, representing the conversion curves of anthracite and coke powder. The remaining curves are experimental and calculated results of MBF and QPF. It can be seen from the figure that they have a high degree of fit. The experimental results are equivalent to the sum of the anthracite combustion conversion rate and the coke powder combustion conversion rate.

It can be seen from Table 6 that the activation energy of the MBF calculated by DRPM is slightly lower than that of QPF. In order to better reflect the height of barriers, Equation (18) was introduced to calculate the apparent activation energy of samples,

$$E_\alpha = c_1 E_1 + c_2 E_2 \tag{18}$$

where $E_\alpha$ is the apparent activation energy of samples in kJ·mol$^{-1}$, $c_1$ is the proportion of the anthracite combustion reaction to the total response, $E_1$ is the activation energy of the anthracite combustion reaction in kJ·mol$^{-1}$, $c_2$ is the proportion of the coke combustion reaction to the total response, and $E_2$ is the activation energy of the coke combustion reaction in kJ·mol$^{-1}$.

The calculation results are shown in Figure 9.

Table 6. Kinetic parameters of samples calculated from different models.

| Simple | Model | $\beta/(°C·min^{-1})$ | $c_1$ | $E_1/(kJ·mol^{-1})$ | $A_1/min^{-1}$ | $\Psi$ | $c_2$ | $E_2/(kJ·mol^{-1})$ | $A_2/min^{-1}$ | $\Psi_2$ | $R^2$ |
|---|---|---|---|---|---|---|---|---|---|---|---|
| MBF | DVM | 5 | 0.3279 | 123.5 | $2.76 \times 10^5$ | - | 0.6721 | 187.3 | $3.20 \times 10^7$ | - | 0.9992 |
| | | 10 | 0.3528 | 137.2 | $2.54 \times 10^6$ | - | 0.6472 | 207.6 | $3.81 \times 10^8$ | - | 0.9993 |
| | | 15 | 0.3023 | 114.1 | $8.32 \times 10^4$ | - | 0.6977 | 175.4 | $6.15 \times 10^6$ | - | 0.9991 |
| | | 20 | 0.3188 | 117.3 | $1.38 \times 10^5$ | - | 0.6812 | 177.6 | $9.14 \times 10^6$ | - | 0.9991 |
| | DRPM | 5 | 0.3279 | 68.6 | 0.028 | $2.74 \times 10^5$ | 0.6721 | 112.6 | 0.434 | $8.27 \times 10^5$ | 0.9994 |
| | | 10 | 0.3528 | 73.1 | 0.105 | $5.97 \times 10^5$ | 0.6472 | 125.8 | 2.318 | $7.51 \times 10^5$ | 0.9995 |
| | | 15 | 0.3023 | 114.3 | $8.32 \times 10^4$ | $8.17 \times 10^{-14}$ | 0.6977 | 107.4 | 0.302 | $3.69 \times 10^5$ | 0.9994 |
| | | 20 | 0.3188 | 117.5 | $13.76 \times 10^4$ | 0 | 0.6812 | 109.8 | 0.433 | $8.34 \times 10^5$ | 0.9995 |
| QPF | DVM | 5 | 0.3078 | 146.6 | $9.60 \times 10^6$ | - | 0.6922 | 221.3 | $2.29 \times 10^9$ | - | 0.9991 |
| | | 10 | 0.2891 | 149.8 | $1.41 \times 10^7$ | - | 0.7109 | 224.2 | $2.80 \times 10^9$ | - | 0.9989 |
| | | 15 | 0.2908 | 140.1 | $3.40 \times 10^6$ | - | 0.7092 | 209.9 | $4.14 \times 10^8$ | - | 0.9991 |
| | | 20 | 0.2926 | 143.4 | $4.64 \times 10^6$ | - | 0.7074 | 212.5 | $5.52 \times 10^8$ | - | 0.9992 |
| | DRPM | 5 | 0.3078 | 67.3 | 0.058 | $1.44 \times 10^5$ | 0.6922 | 130.5 | 4.026 | $9.39 \times 10^5$ | 0.9994 |
| | | 10 | 0.2891 | 81.7 | 0.292 | $8.31 \times 10^5$ | 0.7109 | 136.4 | 6.858 | $9.82 \times 10^5$ | 0.9995 |
| | | 15 | 0.2908 | 92.6 | 0.285 | $2.03 \times 10^5$ | 0.7092 | 125.9 | 2.781 | $9.82 \times 10^5$ | 0.9994 |
| | | 20 | 0.2926 | 73.1 | 0.163 | $5.47 \times 10^5$ | 0.7074 | 98.4 | 0.809 | $2.97 \times 10^5$ | 0.9994 |

**Table 7.** Deviation between the experimental and calculated curves.

| Sample | $\beta/(°C\cdot min^{-1})$ | *RMSE* ($\alpha$)/(%) | | *RMSE* ($d\alpha/dt$)/(%) | |
|---|---|---|---|---|---|
| | | DVM | DRPM | DVM | DRPM |
| MBF | 5 | 1.12 | 0.72 | $3.3 \times 10^{-3}$ | $2.29 \times 10^{-3}$ |
| | 10 | 1.04 | 0.66 | $6.72 \times 10^{-3}$ | $4.62 \times 10^{-3}$ |
| | 15 | 1.12 | 0.74 | $9.37 \times 10^{-3}$ | $6.57 \times 10^{-3}$ |
| | 20 | 1.19 | 0.77 | $1.25 \times 10^{-2}$ | $8.68 \times 10^{-3}$ |
| QPF | 5 | 1.07 | 0.68 | $3.66 \times 10^{-3}$ | $2.52 \times 10^{-3}$ |
| | 10 | 1.17 | 0.68 | $7.65 \times 10^{-3}$ | $4.94 \times 10^{-3}$ |
| | 15 | 1.14 | 0.74 | $1.02 \times 10^{-2}$ | $7.09 \times 10^{-3}$ |
| | 20 | 1.13 | 0.74 | $1.33 \times 10^{-2}$ | $9.24 \times 10^{-3}$ |

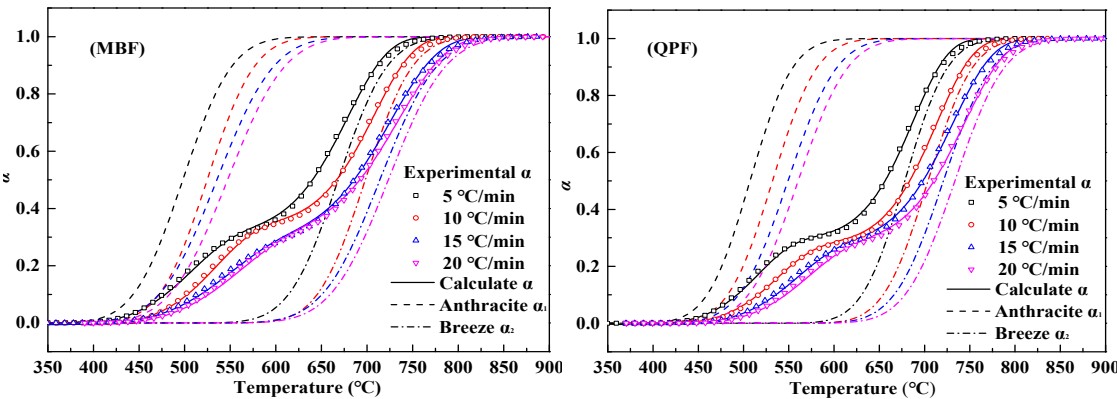

**Figure 8.** Comparison the correlation between experimental data and calculation results of MBF and QPF at different heating rates.

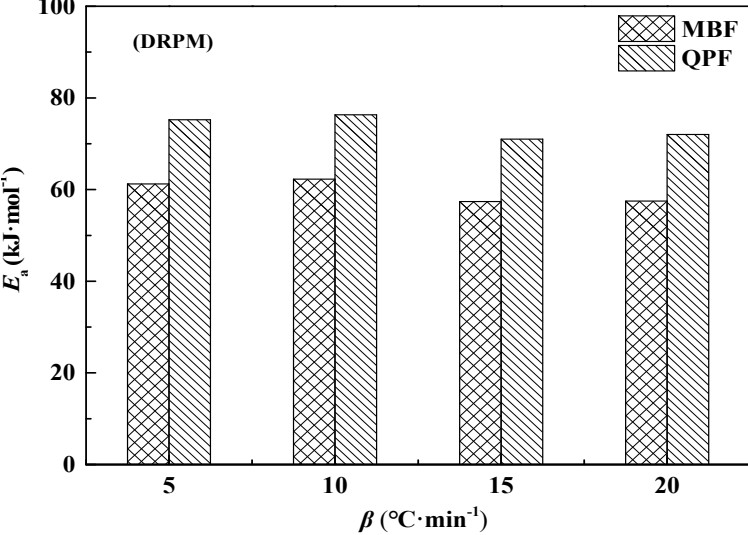

**Figure 9.** Comparison the apparent activation energy of MBF and QPF calculated by DRPM.

At different heating rates, the apparent activation energy of QPF is significantly higher than that of MBF, and the difference between them is maintained between 13.62 and 14.53 kJ·mol$^{-1}$, which does not change with the increase of the heating rate. This indicates that the difference in apparent activation energy between QPF and MPF is the reaction energy barrier provided by the diffusion resistance during the combustion reaction. The inert particles in the quasi-particle structure hinder the contact of the active part on the fuel particles with the air, thereby improving the reaction energy barrier of internal

fuel. Compared with the conventional simple fuel particle structure, an increased number of particles in the quasi-particle structure increase the tortuosity of the internal pores, which greatly reduces the diffusion coefficient of the gas in the heterogeneous reaction and then increases the diffusion resistance of anthracite and coke powder combustion and the activation degree of the diffusion control reaction. However, by comparing the combustion performance parameters of QPF and MBF, it can be seen that the quasi-granular structure can produce a certain heat storage effect on the fuel during the combustion reaction process, which not only improves the combustion performance of the fuel but also increases its combustion rate, thus reducing the adverse effect of diffusion on fuel combustion.

### 3.4. Kinetic Analysis of Quasi-Particle Fuel Combustion

In order to clarify the coupling effect between the redox reaction of iron oxide and the combustion of fuel in the sintering mixture, we carried out SDM in the thermal analysis experiments at different heating rates, as shown in Figure 10. The increase of heating rate accelerates the reaction speed of each reaction in the sintering process and the weight variation of SDM increases with the increase of heating rate. Since the reaction is complicated in the sintering process, a reaction overlap is highly likely to occur. In order to better distinguish the redox reaction and explore the reasons for the change of sample weight, we carried out experiments using a DTA analysis under different heating rates and found that the second derivative of the DTA curve obtained the hidden information of the peak structure in the overlapping region. The results of the analysis are shown in Figure 11.

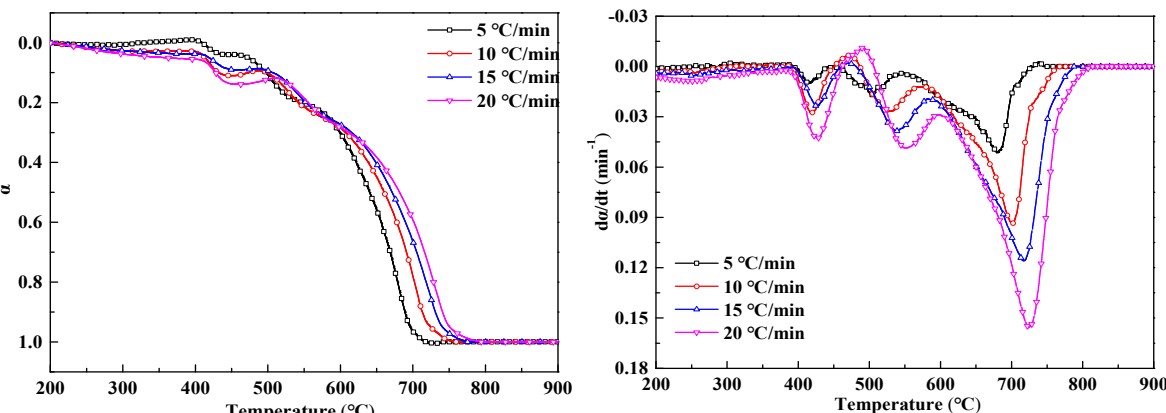

**Figure 10.** Fractional conversion and reaction rate-conversion curves of the sintered mixture at different heating rates.

As can be seen from the figure, SDM exhibits a small change in exothermic → endotherm → exothermic below 500 °C. The volatile first undergoes combustion and exothermic reaction, which provides a reducing atmosphere for $Fe_2O_3$ in the sample and a small amount of CO is sufficient to completely reduce $Fe_2O_3$ to $Fe_3O_4$. Due to the continuous renewal of the reaction gas, the reducing atmosphere is replaced by an oxidizing atmosphere. The $Fe_3O_4$ produced by the reduction is oxidized and the experimental curve shows a slight exothermic weight gain,

$$Volatile + (x/2 + y + z/2)O_2 \rightarrow xCO + yCO_2 + zH_2O \tag{19}$$

$$3Fe_2O_3 + CO = 2Fe_3O_4 + CO_2 \tag{20}$$

$$2Fe_3O_4 + 1/2O_2 = 3Fe_2O_3 \tag{21}$$

The reactions can be regarded as a series of reactions at this stage. Among them, CO and $Fe_3O_4$ are both the product of the former reaction and the reactant of the latter reaction. This indicates that the increase of the latter reaction rate will promote the previous reaction, and the slowing of the previous reaction rate will inhibit the latter reaction, thus forming a significant coupling relationship.

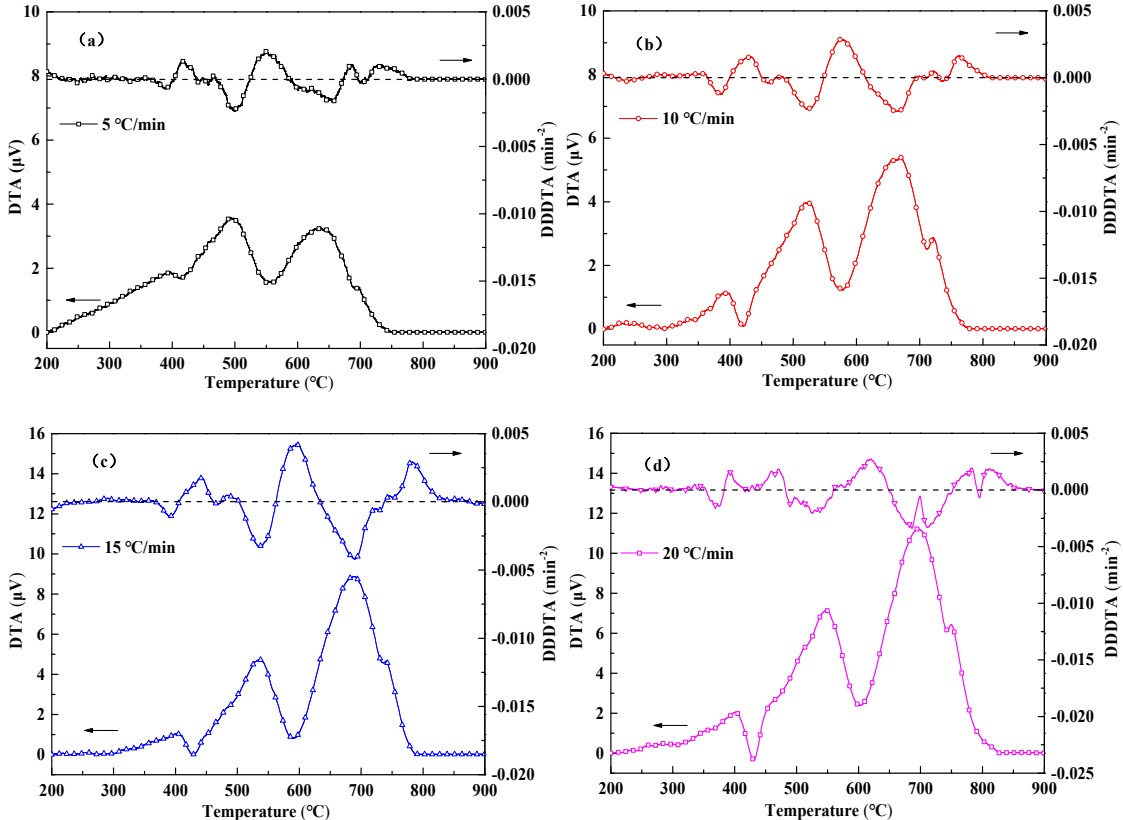

**Figure 11.** Differential thermal analysis curves of the sintered mixture at different heating rates. (**a**) 5 °C/min, (**b**) 10 °C/min, (**c**) 15 °C/min, (**d**) 20 °C/min.

Above 500 °C, anthracite and coke breeze burn, and the exothermic is intense, which may mask the reduction reaction of $Fe_2O_3$. However, the DTA curve shows a significant reduction of the endothermic peak in the 550–630 °C range. This is because this stage is the combustion interval between anthracite and coke breeze and the oxidative exothermic is replaced by the reduced endotherm. However, due to the large weight loss caused by the combustion and reduction reactions, a weight gain phenomenon is not obvious at this stage. The coupling phenomenon of each reaction in the sintering mixture is very obvious and is complicated.

## 4. Conclusions

There are many –CH groups and hydroxyl groups in anthracite, which makes the combustion performance of anthracite significantly better than that of coke powder. When the coke powder is partially replaced by anthracite, the whole combustion process will be accelerated. DVM and DRPM models were used to calculate the dynamics of MBF and QPF at different heating rates. Through comparative analysis of the relevant kinetic parameters and root mean square error, it was found that the DRPM model was more suitable for describing the combustion process of MBF and QPF. Although quasi-particles increase the apparent activation energy of fuel combustion, they also produce a heat storage effect on fuel particles, improve their combustion performance, and reduce the adverse effect of diffusion effect on the combustion reaction process. According to the differential thermal analysis of SDM samples, the coupling between volatiles combustion and redox reaction of iron oxides is obvious in the early combustion period and the oxidation of iron oxides will occur again when the combustion reaction of fuel is weakened.

**Author Contributions:** In this paper, the experimental work and thesis writing were mainly undertaken by J.L. and Y.Y. (Yaqiang Yuan), while J.Z., Z.H. and Y.Y. (Yaowei Yu) were mainly responsible for the review, guidance and revision of the manuscript. All authors have read and agreed to the published version of the manuscript.

**Funding:** This work was financially supported by National Natural Science Foundation of China (51874171) and University of Science and Technology Liaoning Talent Project Grants (No. 601011507-05).

**Conflicts of Interest:** The authors declare no conflict of interest.

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
