# Peer review of "Combustion Kinetics Characteristics of Solid Fuel in the Sintering Process"

_processes, doi:10.3390/pr8040475_

Round 1
Reviewer 1 Report
In the manuscript presented by Jihui LIU et. al treats an important concern that is the use of different solid fuel and fuel mixture, in iron manufacturing. The highest interest is based on the kinetics of the utilized fuels; but are also followed with great interest the fuels properties and their influence in the combustion process.
I think the readers would appreciate an article like this one, but only if they are in the same specific research area.
I consider that the approach is good in term of determination the differences between the coke and the blended fuels – MBF, SDM and QPF. Maybe the cost involvement should be presented comparatively. For these kind of studies the spectral and thermal investigation are fitted and necessary to explain the combustion process and the disparities between the used fuels. The applied models DVM and DRPM establish the most important kinetic data.
The paper is well written with little flaws in typing or spelling:
2nd page at Materials - for the particle size of iron oxide and alumina are given negative values
Fig 4 – the x axis is in Chinese?
Fig. 11 for the DTA curves is not clear what samples are presented in what order and the graphic is not well aligned.
The drawn conclusion are sustained of the information presented in the manuscript.
The References are sloppy written, should be revised.
Reviewer 2 Report
To the editor,
Thank you for giving me the opportunity to review “Combustion Kinetics Characteristics of Solid Fuel In Sintering Process" for publication in Processes. The authors studied solid fuels commonly employed in iron production. Fuel properties and its combustion characteristics are examined for the influence of three factors. Homogenous anthracite and coke powders as well as three blended fuels (MBF, QPF, and SDM) are examined. Spectral and thermogravimetric measurements were performed to study fuel properties and their combustion characteristics. DVM and DRPM models are devised to determine relevant kinetic parameters. The subject matter is relevant to the journal, and will be of interest to readers. The manuscript is well laid out and well presented. Overall, I recommend the publication of the manuscript with minor revisions as listed below:
1. A general description of how these fuels are produced would help readers that may not be intimately familiar with solid fuel combustion.
2. The authors compared 3 different blended fuels. The first figure would seem to indicate that the fuel particles are modeled as perfect spheres. Were any other surface morphology for fuel particles considered? How relevant is surface morphology to combustion characteristics?
3. The authors stipulates that the QPF flammability index, combustion characteristic index are significantly higher that MBF (page 7). They further note that this indicates that the quasi-particle structure reduces the reaction ability of the fuel in the initial stage of combution. However, this seems to contradict a previous statement by the authors in page 2 where they mention that larger combustion characteristic index indicate better combustion performance. Also in the same paragraph in page 7, the authos state that a "regenerator" is formed around the wrapped fuel. Has this behavior been observed in a model or an a posteriori analysis? If so, how was it modeled or observed?
4. Is there a specific reason why REDOX is always capitalized? Also particles sizes given in page 2 are negatives.
5. Figures: Figure 4 x-axis label is in chinese; Fig 4,5,7,8,10 are difficult to read, especially the markers as they are so very small and difficult to percieve when printed black and white. Insets in figure 5 and 10 are very difficult to read and would recomend that they are made into their own figures.
Thank you
Reviewer 3 Report
The manuscript is about combustion kinetics characteristics of soild fuel in sintering process. The scope of this article is consistent with the requirements of the Processes journal, but it requires major revision in accordance with the comments below:
- Avoid lumping references as in ([1-6], [12-14], [19-23]). Instead summarise the main contribution of each referenced paper in a separate sentence.
- I think that the name of point 2 should be Materials and methods.
- What values of absorbance are on the figure 3?
- On the figure 4 is wrong language font.
- On the page 9 is the sentence: “The calculation results are shown in Fig 7.” and should be “The calculation results are shown in Fig 9.”
- Figure 11 presents 4 graphs: a, b, c, d but it is not written what they mean.
- References are not prepared in accordance with the journal's guidelines.
- It would be good to cited similar articles from the Processes journal.
Round 2
Reviewer 3 Report
The authors have made all the corrections. The manuscript may be published in its current form.
Author Response
Dear editor,
Thank you for your letter and for the reviewer’s comments concerning our manuscript entitled“Processes-740589”. Reviewer 3 think that we have made all the corrections. The manuscript may be published in its current form.
Academic Editor comments are all valuable and very helpful for revising and improving this paper. We are revising these issues.
Thanks for your comments and suggestions.
Best regards,
Ms. Jihui Liu